# Marina Observation of Sea Turtles: Establishing a Database of Intracoastal Waterway Green Sea Turtles in Northeast Florida

**DOI:** 10.3390/ani13020279

**Published:** 2023-01-13

**Authors:** Edward McGinley, Avery Cogley, Leslie Palmer, Patricia McCaul, Lilli Longo, Jasmine Silvennoinen, Angela Martin, Josalyn Gomez, Sydney Bachmaier, Michaela Mackey, Chris Kao, Scott Eastman, Catherine Eastman

**Affiliations:** 1Department of Natural Sciences, Flagler College, St. Augustine, FL 32084, USA; 2Department of Mathematics and Technology, Flagler College, St. Augustine, FL 32084, USA; 3Office of Resilience and Coastal Protection, Florida Department of Environmental Protection, St. Augustine, FL 32080, USA; 4Whitney Laboratory, University of Florida, St. Augustine, Fl 32080, USA

**Keywords:** juvenile green sea turtle, intracoastal waterway, head scales

## Abstract

**Simple Summary:**

Green sea turtles are currently listed as an endangered species. Therefore, any information on unstudied populations could provide valuable insight into the status of the species. Within the intracoastal waterway in northern Florida, juvenile green sea turtles are consistently seen eating the large leafy algae growing on the local marinas. The goal of this project was to use photographic records of each green sea turtle to determine whether individuals could be identified at two of these local marinas. Unique individuals were successfully identified, which allowed us to keep records of how often and where we saw certain turtles. To date, we have identified 195 turtles over an 18-month study period. The largest numbers were documented in warmer summer months, and the numbers decreased in colder winter months. Individual turtles were seen anywhere from once to 26 different encounters, with the longest duration between the first and last sighting of 569 days. This information can help bolster support for enacting local regulations to help conserve juvenile green sea turtles.

**Abstract:**

As conservation efforts regarding green sea turtles, Chelonia mydas, continue, it is imperative to document behaviors and foraging habits/habitats of understudied populations. We have conducted an 18-month study dedicated to photographing the local population feeding alongside floating docks within the Guana Tolomato Matanzas estuary to determine the capability of matching head scale patterns efficiently through a pattern matching program: HotSpotter. To date, 195 unique sea turtles have been identified between two different marinas located in St. Augustine, FL. Of these, 98 were spotted more than once, with 39 of them being “tracked” for longer than a year. Temperature trends were also monitored in conjunction, showing that more individuals appeared during the warmer months of the year. The evidence, overall, indicates that these locations host a resident population of green sea turtles, leading to the need for a discussion on potential threats originating from the usage of these marinas by humans.

## 1. Introduction

All species of sea turtles are currently listed as endangered or threatened [1], including the green sea turtle. Assessments have shown that several green sea turtle populations are no longer declining since the 1978 Endangered Species Act was put in place, and in fact, some may be recovering [2,3]. However, it is also important to note that Casale & Ceriani [4] indicate that sea turtle populations overall may be overestimated because they are usually determined from annual nest counts. Specifically, nest counts have the potential to be unreliable as an overall recovery indicator due to inconsistent reproductive rates of the females. This is because female adult sea turtles have the ability to lay multiple nests in one year but may also not lay a nest every year [4,5]. Additionally, nest counts cannot accurately determine the mortality or survival rates of male and juvenile sea turtles [6]. Due to the inability to accurately estimate population size based solely on nesting behavior, it is important to better study various life stages of the different sea turtle species as well as account for the various populations within a species and critical foraging areas.

Once green sea turtles have left the nest, they can be hard to track due to the “lost years” in the Sargasso Sea and their migratory behavior [7,8,9,10]. When these turtles reappear as juveniles and adults, the density of the populations can vary depending on location [7]. Areas where humans are scarce tend to show denser populations [7]. Such distributions may be due to anthropogenic factors as a main cause for mortality in green sea turtle populations [11] as well as multiple non-lethal anthropogenic and abiotic threats, including loss of habitat and nesting beaches, infectious disease [12], skewed sex ratios due to climate change [13], and dietary shifts related to warmer temperatures [14]. Therefore, it is critical to fill knowledge gaps in understudied populations of green sea turtles.

For example, there is a lack of information regarding the population size in northeast Florida waters and how long individuals might reside here. In order to fill this knowledge gap, data needs to be collected with regard to tracking individual sea turtles. There is a multitude of ways to track green sea turtles. Satellite tagging [8,15], passive integrated transponder (PIT) tagging [16], isotopic tracking [17,18], metal flipper tags [16], plastic tags [9], acoustic tagging [19,20], genetic tags [21] and photo identification (PID) [22,23,24] have all been used. Plastic tags have previously been known to be unreliable as they will fade and become unreadable over time. Additionally, some metal tags can cause negative reactions with sea turtles and also have the potential to detach from them and become lost [16]. PIT tags are more reliable and efficient but require recapture to track turtles and their migratory behavior [16]. Blood samples [9] and tissue samples [17] have filled some gaps in the distribution and migratory behavior by identifying foraging grounds through stable isotope analysis. However, all of these methods of tagging require handling the sea turtles, unlike PID [22].

PID is used to track individuals in a much less invasive manner compared to various physical tags [25,26]. Any physical tag used always carries the possibility of being shed by the organism [16,26]. With sea turtles, identification can be performed using a variety of areas: facial scutes [25,26,27], flipper patterns [24,28], and carapace patterns [29]. The photos can be used to differentiate one individual from another within the same population, similar to human fingerprints [22,23,25,30]. Carpentier et al. [25] showed that the scale patterns are also stable over time so that PID can be used to track individuals through multiple years.

There are several PID programs available, such as APHIS [24], I^3^S [31], and HotSpotter [22,23,25,30]. Focusing on HotSpotter, it works by identifying and matching the pattern of unique features to help identify similarities, or lack thereof, between two different photos [22,30]. Previously, it has successfully identified individual animals within populations of giraffes, jaguars, and lionfish with exceptional results [22]. Within the sea turtle community, Dunbar et al. [30] had success in using the program with hawksbill sea turtles in Honduras. This was achieved by photographing the dorsal head scale patterns of those turtles and using this program to compare them to previous photos taken and added to a database. As we know that green sea turtles can also be uniquely identified based on their facial scale patterns, it is likely that HotSpotter would be successful in matching individuals within their population as well [25,26].

Green turtles have been observed regularly inhabiting inshore waters around St. Augustine, Florida, USA, including foraging for macroalgae found in and around marinas (pers. obs.). Such proximity to frequent vessel traffic poses a significant threat to individual turtles in the form of vessel strikes, particularly considering the turbid nature of these inshore waters [32,33]. Understanding population size and residency patterns are therefore crucial for conservation management decisions for this turtle population. The first objective of this study was to determine if HotSpotter can be used to identify green sea turtles from photographs taken while the observer is on the dock and the turtle is under the surface. The second objective was to determine individual recurrence and ascertain if they remain at a marina for extended periods of time or if they are more transient. Knowing where these turtles are feeding can lead to better conservation of their habitat, as well as further studies on what they are eating and their population dynamics which can allow for a greater understanding of their life history [34,35].

## 2. Materials and Methods

### 2.1. Study Sites

Two marinas were selected for this study located in St. Augustine, FL: the Conch House (CH) and Camachee Cove (CC) (Figure 1). Both marinas in this study were floating concrete docks secured to concrete pilings. The CH’s perimeter measures 4.4 km and CC’s measures 4.8 km. Both marinas are privately owned and have several hundred slips, 194 for CH and 260 for CC (Figure 2). These marinas support private boaters, head boats, and ecotour operations.

### 2.2. Surveying Protocol and Photo-Capture

Undergraduate students were selected and trained on the proper protocol for this project by the PI (Dr. McGinley). Surveys were conducted weekly, on weekdays during the volunteer’s free time (~120 minutes per survey). Surveys were limited to weekdays to accommodate a request from the CH harbormaster to avoid surveys on the weekend due to crowds from the restaurant on the premises. No surveys were conducted on weekends at CC to keep this aspect consistent. These walks occurred from June 2020 through December 2021, for a total of 576 days. New volunteers added to the project throughout its duration were required to go with an established volunteer (or Dr. McGinley if no experienced volunteer was available) to learn how to handle the camera and observe the turtles. 

Because of these constraints, it necessitated having walks on different days and different start times at the two marinas. As we were not looking to directly compare the marinas but rather to look at green sea turtle abundance and residency at the different marinas, a non-standardized start time and day was deemed acceptable.

Volunteers started at the same point at the marina each time and were instructed to walk along the edge of the dock and search for green sea turtles until the entire perimeter was searched. When a turtle was encountered, students were instructed to slowly approach the turtle and take pictures of the top of the head and, if possible, the lateral head scale patterns. All photos were taken with either a Canon EOS Rebel T7 or a Canon EOS 4000D camera equipped with either an 18–55 mm and 75–300 mm lens.

Due to the water clarity at both of the marinas, turtles were able to be photographed while they remained under the water but at the surface, either feeding or moving. The act of feeding often required the turtle to tilt its head, which allowed the lateral head scales to be captured. 

Missed turtles were not documented until March 2021. In the event that a turtle swam away before photos of the head could be taken, a period of five minutes was timed before the encounter was registered as a “miss”. At this point, a picture of the water was taken to indicate that a sea turtle sighting was missed. 

All photos were uploaded to the database and organized by date, location, turtle number, and profile direction. The photos were cropped and rotated depending on the angle they were taken at. If the photo showed only the top of the turtle’s head, the photo was rotated so that the nose was facing in the upward direction. For the lateral pictures of the head scales, photos were oriented facing left or right.

### 2.3. Photo-Identification

HotSpotter was selected to assist with the identification of unique individuals [30]. A region of interest was created manually in the photo by the researcher and became a “chip”. A query was run based on this chip, comparing it to all previous chips in the database. 

HotSpotter then provided six chipped photos to compare against along with a similarity ranking (0–6; 0 = most similar, 6 = least similar) to help determine the likelihood of two different photos originating from the same turtle (Figure 3). Red and yellow vertical lines indicate similar keypoints in both photos. A manual selection was made to officially determine whether this turtle has been seen before or if it is a new and unidentified individual.

### 2.4. Statistical Analysis

Summary statistics regarding the number of unique individuals, sighting frequency, and spatiotemporal patterns were investigated using the stats program R [36].

In addition to observer sighting data, water temperature data were available from a submerged data sonde (YSI EXO2 data sonde) maintained by research staff at the Guana Tolomato Matanzas National Estuarine Research Reserve. This multimeter measures temperature, along with specific conductance, salinity, dissolved oxygen, pH, depth, and turbidity. The data are stored on a publicly accessible site (https://cdmo.baruch.sc.edu/, accessed on 6 November 2022), and all temperature data were downloaded from there. 

## 3. Results

We surveyed the two marinas a total of 153 times (CC: n = 75; and CH: n = 78) over 576 days. Hotspotter analysis identified a total of 195 unique individuals from 2840 available photos. We recorded 354 green turtles at CC during the project, and 87 of these turtles were unique individuals (24.6%). At CH, 323 turtles were documented, and 108 of these turtles were unique individuals (33.4%). The turtles seen during this project had sightings ranging from one encounter (n = 97; Figure 4) through 26 encounters (n = 1).

Of the unique turtles seen, 39 of the turtles were “tracked” for over one year (20.0%; Figure 5). Eighty-five turtles (43.6%) were tracked for at least one month. The longest tracking duration for a turtle was 569 days. 

There was a distinct difference in the duration of tracking between the two different marinas (Figure 6). The average duration between the first and last encounter was 86 ± 156 days (µ ± SD 0–326 days) for the entire project. The tracking duration was 136 ± 196 days at CC and 54 ± 113 days at CH. Both marinas had a mode of 0 days, indicating that most turtles are only seen once. However, the plot also indicates that turtles at CC are much more likely to be tracked for longer durations, i.e., several hundred days, as well as having the lengthiest durations at CC, i.e., 450+ days.

The cumulative number of turtles shows a constant trend after five months at CC and seven at CH marina (Figure 7). An increase in new turtles was seen at the CH marina in July and August 2021 before the numbers started to stabilize again; these large increments are not observed in CC.

Water temperature within the San Sebastian River ranged from a high in July/August 2020 (29.2 ± 0.8 °C; Figure 8) to a low of 17.7 ± 1.2 °C in January 2021. More turtles were encountered, on average, during the warmer months, with a peak at CH in May 2021 of 13 ± 0.6 turtles and 13 ± 3.6 turtles at CC in July 2020. Turtles were seen in all 38-month and marina combinations (19 months × 2 marinas).

## 4. Discussion

HotSpotter proved reliable for the identification of individual green sea turtles within the murky water of the Guana Tolomato Matanzas (GTM) estuary. This capability of HotSpotter and PID created a faster, streamlined process of identifying and “recapturing’’ green sea turtles without handling them [37]. By using this software, we created a photo database of 195 unique green sea turtles. Secondary findings show a population of green sea turtles which resides at the local marinas allowing us to study their reoccurrence, the longevity of an individual residing in the area, and variation in their appearance based on seasonal temperature.

In this study, we were able to track turtles for a year and a half (576 days). During the 18 months of this study, 51.3% of turtles (98 out of 195) were seen more than once, with only two individuals seen moving between marinas (one being a previous patient at the Sea Turtle Hospital at Whitney Laboratories (University of Florida) between sightings). All other individuals were only re-sighted at the same marina where they were originally found. Because of the close proximity of the marinas, it would be difficult to differentiate them as distinct habitats due to the large range of juvenile green sea turtles: from 0.7 to 5 km^2^ on shallow reefs in Palm Beach, FL [38] to 374–2060 km^2^ in the Everglades in Florida [39]. With monthly average sightings above zero, we know sea turtles remain in this area year-round. The counts did decrease as the temperature dropped, indicating that some turtles may be leaving this area [40], resting more frequently [16,40], or moving into deeper water [41]. Those who decide to stay become more susceptible to being cold-stunned and in need of medical attention [41]. Studies such as this provide an opportunity to investigate potential relationships between the number of turtles seen at the marinas and the number of patients admitted during the colder months to local sea turtle hospitals.

The number of new sea turtles seen at each marina leveled off during the 18 months of this study, except for a jump from August to September 2021 at Conch House. This could allow more fine-scale observation of habitat use and patterns of immigration. Typically, efforts to estimate population size separate from nest counts usually require time- and labor-intensive methods as well as a potentially high cost: sightings from ships [42,43], towed diver surveys [44,45], and aerial surveys [46,47]. Recently, the use of unmanned aerial surveys (UAV) has proven to be a reliable and cheap alternative [43,48]. UAVs are able to fly lower than planes and do not potentially disturb turtles as a boat or diver might. Unique information can be gathered on not only abundance estimates but also behaviors [48].

The idea behind the UAV surveys is very similar to the methodology in this project, except instead of unmanned aircraft, the imagery is captured by a handheld camera. Because the green turtles are habituated to people because of the daily foot traffic at the marina, we are able to observe the behaviors of juvenile green turtles. The unique markings of each turtle will allow the possibility of recording behaviors of specific individuals related to feeding, response to the presence of people, as well as any intraspecific interactions.

The rationale for such persistent use of local estuaries and marinas, particularly among juvenile turtles, is an important factor relating species distribution and conservation management. Bolten [49] looked at loggerhead sea turtles that recruit from an oceanic development stage to a neritic one and found a steeper decrease in growth rates (cm/yr) for loggerheads as they grew in an ocean environment. By including a life stage that involves shallower water, loggerheads were able to maximize growth potential versus remaining in a solely oceanic habitat. Green sea turtles follow a similar oceanic-neritic life history. Several studies have noted the inshore size of green sea turtles ranges from 30 cm to 50 cm along their straight carapace length (SCL) while studying the Indian River Lagoon, FL, USA [50] and Palm Beach, FL, USA [38]. Hart and Fujisaki [39] noted a similar minimum size but a higher maximum size (67.5 cm) for a population of green sea turtles in the Everglades, FL, USA. By comparison, the Sea Turtle Hospital at Whitney Laboratories has reported SCL ranges from 25.5 cm to 51.8 cm for this estuary [51].

Foraging opportunities likely play a large role in such ontogenetic shifts and in specific habitat selection. Studies conducted in Texas mention patchy seagrass distributions in their estuaries and found sea turtles feeding on macroalgae growing on hard structures such as jetties [52,53]. Even when there was equal access to different habitat types, Chambault et al. [54] found that habitats were not used equally and that individuals showed contrasting responses to similar conditions. This is a similar theme in other studies, showing that green sea turtles have a large degree of intra-species variability in diet [55,56]. Because of these differences, it is important to identify unique foraging areas, as these data will help determine potential risks to sea turtles as it relates to plastic ingestion [57,58], exposure risk to HAB toxins [59,60], boat strikes [61,62], and other deleterious effects that could be influenced by the location of the foraging area.

Adding literature on unstudied foraging areas, such as man-made structures, can contribute to the knowledge and understanding of green sea turtles in a largely anthropogenically affected world. We were able to observe green sea turtles foraging on macroalgae growing on floating dock platforms during the day. Due to the high human traffic, these floating docks may need conservation efforts and community awareness to protect this local population from any negative anthropogenic impacts. Boats are constantly leaving and docking at marinas and can pose a threat to many marine species, including green sea turtles [32,33]. Several of our locally studied turtles presented evidence of injuries related to boat strikes. In addition, there is a risk of leakage of oil as boats fuel up at these marinas. Wallace et al. [63] note the possibility of negative genetic impacts on the female’s eggs after ingesting the chemical components of oil spills. 

Furthermore, nest counts are currently the dominant statistic determining trends and population sizes of sea turtles. However, there are well-documented challenges to using such methods, including the lack of information on marine life stages [4] Long-term studies, especially for young sea turtles, are difficult to conduct but necessary for accurate population assessments and conservation planning. Hence, additional literature is necessary for local populations to help determine their status and if additional efforts to conserve the area are required.

## 5. Conclusions

This study shines light on the large number of turtles sighted in only two small areas of St. Augustine, FL, over a short period of time. The GTM estuary, and more specifically, man-made structures such as marinas, may serve as important foraging grounds for juvenile green sea turtles. 

A project using photo-identification like this lends itself easily to citizen science. The ubiquity of smart devices has allowed most people to carry a high-powered camera in their pocket. This fact, coupled with these marinas’ high foot traffic, makes an ideal situation for public inclusion in our dataset. We are currently working on a protocol that involves proper permitting, the cooperation of the harbormasters and the general public, and Flagler College to safely and constructively establish a citizen science project monitoring green sea turtles at local marinas.

In addition, further community awareness and involvement have proven to help the conservation efforts in other areas and are an important next step here as well [37]. Community and legislative awareness coupled with consistent scientific monitoring can work in tandem to conserve and protect this charismatic and ecologically impactful species and consequently aid in bringing about an umbrella effect to maintain this paramount but increasingly fragile estuarine ecosystem.

## Figures and Tables

**Figure 1 animals-13-00279-f001:**
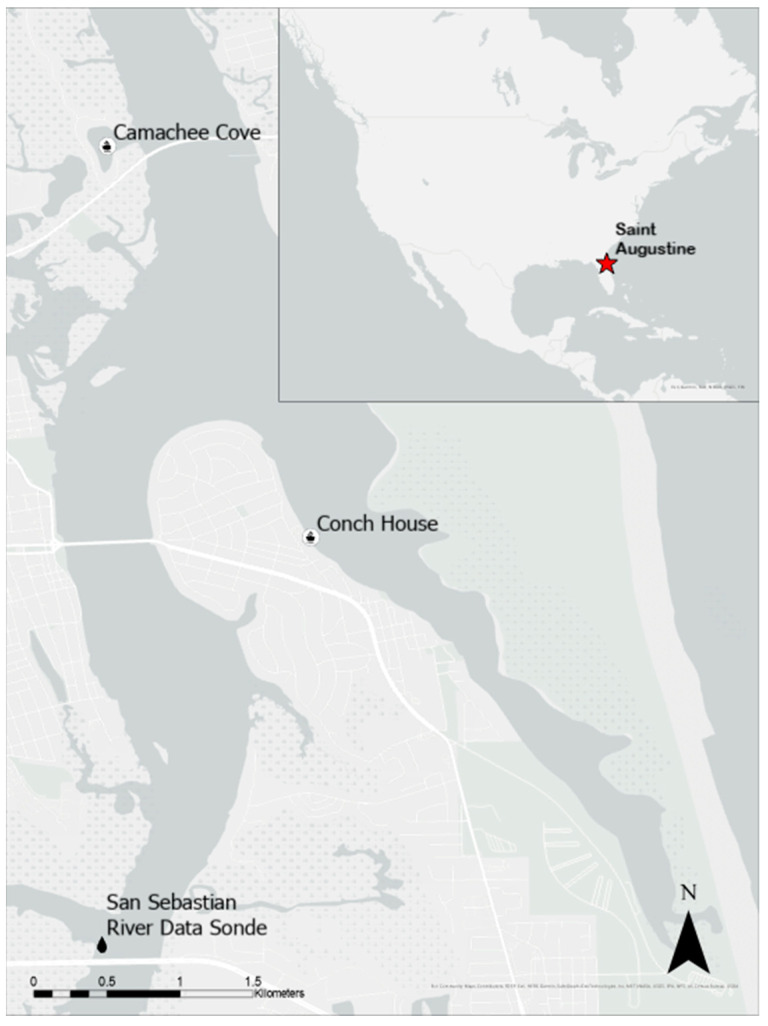
Study sites for this project and the data sonde where water temperature was recorded. This study took place at two marinas located in St. Augustine, FL on the Atlantic coast of the US.

**Figure 2 animals-13-00279-f002:**
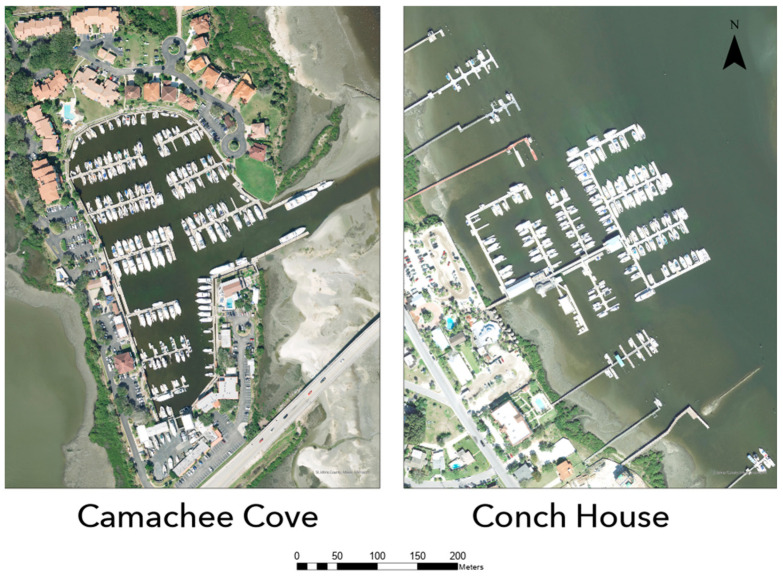
Satellite imagery depicting closeups of both study sites. The Conch House Marina only includes the interconnected structure in the center.

**Figure 3 animals-13-00279-f003:**
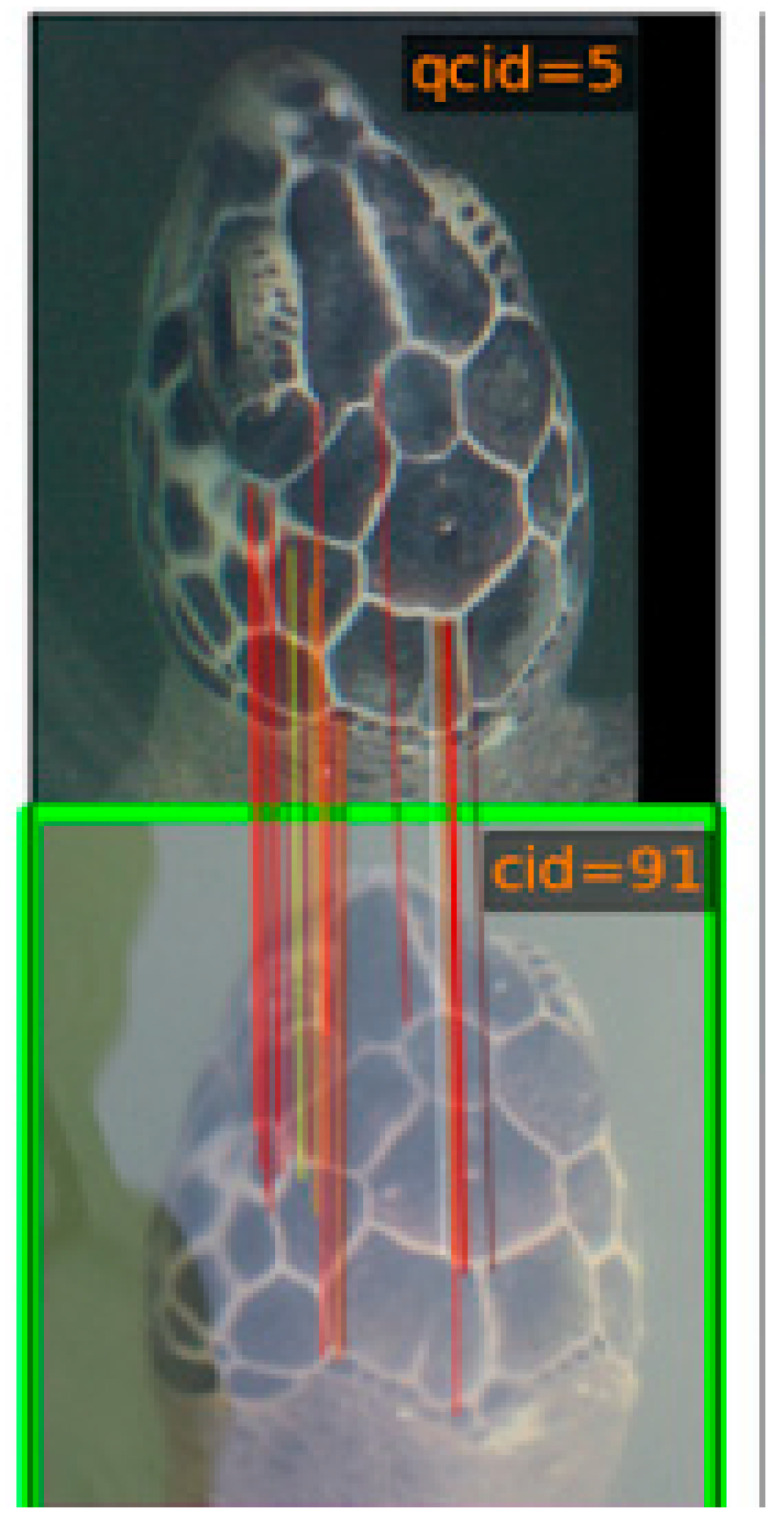
Output from HotSpotter. The program uses a matching algorithm to determine the most likely matches within a photo database. Above is an example of an unknown turtle (**top** picture) being matched with a turtle already in the database (**bottom** picture).

**Figure 4 animals-13-00279-f004:**
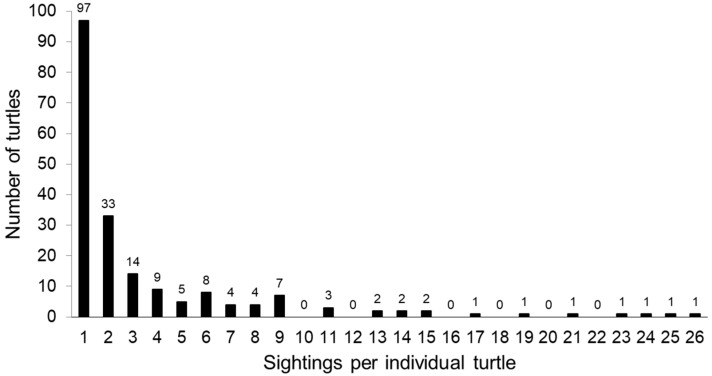
The number of sightings per turtle. Of the 195 identified, 97 were only seen one time. The maximum number of sightings was 26 for one turtle. The number above the bars indicates the sample size.

**Figure 5 animals-13-00279-f005:**
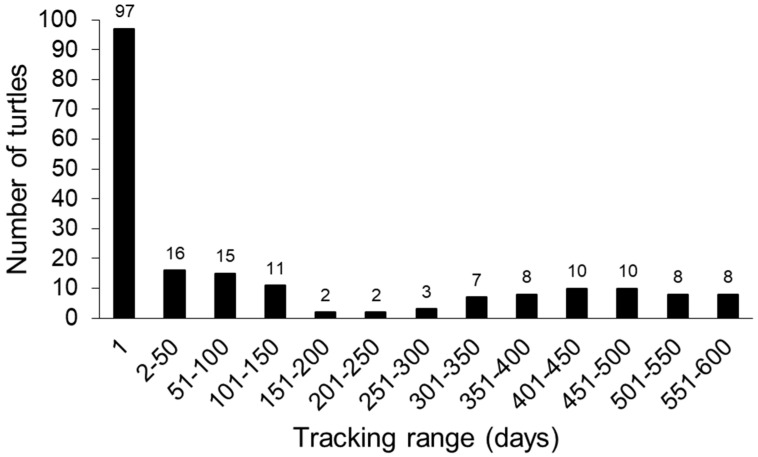
Tracking duration for each turtle in the project. Turtles seen only once were designated as being tracked for one day. The number above the bars indicates the sample size.

**Figure 6 animals-13-00279-f006:**
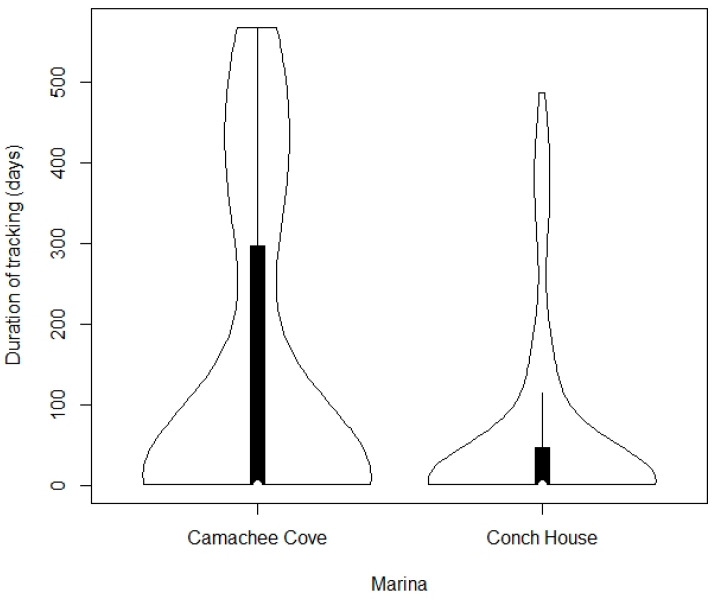
Violin plot of the tracking duration separated by marina.

**Figure 7 animals-13-00279-f007:**
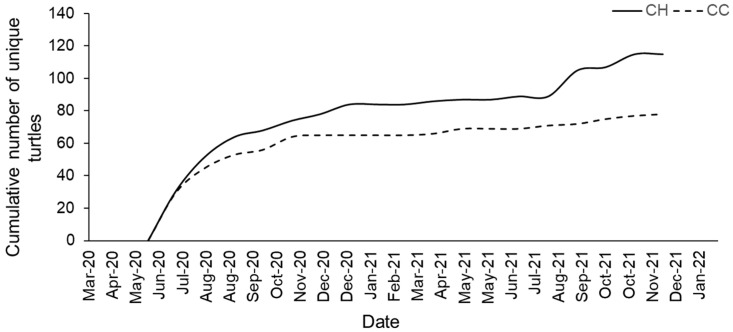
The cumulative number of turtles for Conch House (CH) marina and Camachee Cove (CC) during this study period.

**Figure 8 animals-13-00279-f008:**
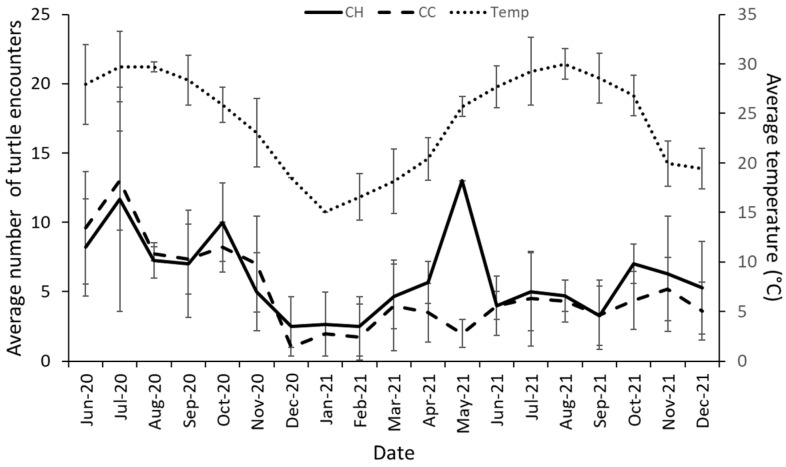
The average monthly sightings for each marina. The Conch House Marina (CH) was visited 78 times, while Camachee Cove (CC) was visited 75 times. The monthly average temperature for each month was used from a NERR data sonde located near both marinas. All data points represent the mean ± standard deviation.

## Data Availability

The data presented in this study are available on request from the corresponding author. The data are not publicly available due to the large amount of memory required to store the photos. Photos are now housed on the Internet of Turtles website, a Wildbook repository (account required) at the following link: https://iot.wildbook.org/.

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
