# Peer review of "Marina Observation of Sea Turtles: Establishing a Database of Intracoastal Waterway Green Sea Turtles in Northeast Florida"

_animals, 2023, doi:10.3390/ani13020279_

Round 1
Reviewer 1 Report
The study shows a methodology that could be applied in other locations to identify other key areas for sea turtles. However, the manuscript needs to improve it. Comments and recommendations are included in the document.

Author Response
I combined all comments into one Excel sheet and separated it by reviewer and editor. We did notate if we did changes on the sheet but all corrections were made in the Word document itself. I will email the editor a comment of the Excel sheet log.

Reviewer 2 Report
Methods
In this section, the information in the study area is a bit confusing. I suggest rewriting this part and making it more clear to fully understand how the surveys were done, how big was the sampling area, and how many kilometers or meters were walked. Also, several methods are written in the results section. An example of this is the delimitation of the sampling period, in methods, it is stated when the surveys started but not when they finished; it is stated later on in the results. As well as the statistical analysis is fully explained in results when they should be explained and described in methods (paragraph with lines between 235 to 249).
My recommendation is to properly separate both sections, everything that the authors use to respond to the objectives of the study should be explained and described in methods, then, whatever came out of that should be in the results section.
Results
Page 7, lines 228 to 231, my suggestion is to rewrite it. Here is a suggested change: ''The cumulative number of turtles shows a constant trend after five months at CC and seven at CH marina (Figure 6). An increase in new turtles is seen at the CH marina in July and August 2021 before the numbers started to stabilize again; these large increments are not observed in CC''.
Also, there are several times that the correct symbol for the temperature is not written (only the C without the degrees symbol), the same as in figure 5. Figures 5 and 6 are missing the X-axis titles.
Conclusions
There aren't any conclusions on the study in this section, only recommendations about the next steps. Authors should write in this section whether or not they were able to answer the main objectives of the study, writing the main highlights of their results and then, how they can use them or how the results can help with conservation plans and regulations in these types of areas.
References
Line 381 - the scientific name is not written correctly
Ref 24 - the scientific name is not written correctly (missing capital letter in Eretmochelys)
Refs: 9, 12, 13, 14, 15, 20, 24, 25, 29, 30, 31, 32, and 33 have the scientific name in normal typing. They should be in italics or have the name underlined (it will depend on the rules of the journal).
Author Response

(The authors gave the same response as above.)

Reviewer 3 Report
I enjoyed reading your manuscript and really liked the use of HotSpotter to identify individual green sea turtles. It is definitely something that could be used globally if a researcher wanted to do a similar study in their area.
I do think the manuscript needs major revisions. I have included an annotated copy of the manuscript with my comments throughout. I have questions regarding missing methodology as well as some of the analysis. I think you were missing some key details that readers need to know to understand what was done and duplicate the study.

Author Response

(The authors gave the same response as above.)

Round 2
Reviewer 1 Report
The manuscript has improved considerably. Only minor grammar checks are necessary.
Author Response
I have addressed the editor's comments

Reviewer 3 Report
This manuscript is greatly improved from it's original version. It is obvious the authors took the reviews comments into consideration when revising. I recommend this manuscript for publication.
Author Response
I have addressed the editor's comments.
